# Semi-Supervised SAR ATR Framework with Transductive Auxiliary Segmentation

**Chenwei Wang, Xiaoyu Liu, Yulin Huang \*** [ID]**, Siyi Luo, Jifang Pei** [ID]**, Jianyu Yang** [ID] **and Deqing Mao**

School of Information and Communication Engineering, University of Electronic Science and Technology of China, Chengdu 611731, China
* Correspondence: yulinhuang@uestc.edu.cn

**Abstract:** Convolutional neural networks (CNNs) have achieved high performance in synthetic aperture radar (SAR) automatic target recognition (ATR). However, the performance of CNNs depends heavily on a large amount of training data. The insufficiency of labeled training SAR images limits the recognition performance and even invalidates some ATR methods. Furthermore, under few labeled training data, many existing CNNs are even ineffective. To address these challenges, we propose a Semi-supervised SAR ATR Framework with transductive Auxiliary Segmentation (SFAS). The proposed framework focuses on exploiting the transductive generalization on available unlabeled samples with an auxiliary loss serving as a regularizer. Through auxiliary segmentation of unlabeled SAR samples and information residue loss (IRL) in training, the framework can employ the proposed training loop process and gradually exploit the information compilation of recognition and segmentation to construct a helpful inductive bias and achieve high performance. Experiments conducted on the MSTAR dataset have shown the effectiveness of our proposed SFAS for few-shot learning. The recognition performance of 94.18% can be achieved under 20 training samples in each class with simultaneous accurate segmentation results. Facing variances of EOCs, the recognition ratios are higher than 88.00% when 10 training samples each class.

**Keywords:** synthetic aperture radar (SAR); automatic target recognition (ATR); semi-supervised; training loop process

## 1. Introduction

Synthetic aperture radar (SAR) is an important microwave remote sensing system in military and civil fields, and SAR automatic target recognition (ATR) is one of the most crucial and challenging issues in SAR applications. Recently, many outstanding scholars have proposed diverse deep learning-based methods and achieved remarkable results in SAR ATR applications [1–5].

However, many of these algorithms require that the training set consists of abundant labeled samples for each target type, which is often impossible to be satisfied in practical applications. Furthermore, under some conditions, e.g., earthquake rescue and sea rescue, the number of obtained SAR images can be only a few, and these existing SAR ATR methods probably lose their effectiveness. This has led to studies on few-shot learning (FSL) [6–8], which builds new classifiers from very few labeled SAR images, and these studies also continue to drive towards efficient and robust SAR ATR.

Previous approaches of FSL in SAR ATR mainly can be divided into three categories: data augmentation methods, metric-based methods, and model-based methods. The data augmentation methods improve the recognition performance by enriching the amount and enhancing the quality of training SAR images. For example, Zheng et al. proposed a semi-supervised SAR ATR method via a multi-discriminator generative adversarial network (GAN), which achieved an 85.23% recognition rate with 20 samples for each target [9]. Gao et al. proposed a semi-supervised GAN with multiple generators, which achieved more than 92% with around 40 samples in each target type [10].

The metric-based methods are based on learning representations for classes that can generalize to new classes. For example, Li et al. proposed a conv-biLSTM prototypical network, which has a classifier based on the Euclidean distance between training samples and the prototypical of each class [11]. Fu et al. designed a hard task mining method for effective meta-learning which yields better performance with the largest absolute improvements of 1.7% and 2.3% for 1-shot and 5-shot, respectively [12]. The model-based methods use prior knowledge to construct the embedding space and constrain the complexity. For example, Persello et al. proposed an active and semi-supervised learning network for SAR ATR, which is a semi-supervised support vector machine (SVM) [13]. Li et al. proposed a classification approach that adopts the canonical scattering models widely used in model-based decompositions to provide an improvement for the well-known $H/\alpha$ classification [14].

However, the challenge of FSL in SAR ATR is whether to have a helpful inductive bias [15–17], which can improve performance on novel classes but is hard to develop when the feature embedding has a large difference between a few samples and full samples [15,18]. In essence, these methods above just further mine the few labeled SAR images to obtain more usable recognition information. During the process, these methods cannot acquire new information from these few samples in terms of information theory, which hinders the improvement of recognition performance in SAR images.

In the light of deep segmentation methods for SAR images [19,20], the target segmentation has become more achievable and can reveal the unique morphological features that can be extended for use to SAR ATR under a few labeled training SAR samples. For example, Ref. [21] proposed a differentiable superpixel generation method and a superpixelwise statistical dissimilarity measure method with the encoder–decoder structure. Ref. [22] proposed one differentiable boundary-ware clustering method and the soft association map with an encoder–decoder structure. Meanwhile, the segmentation of SAR images is also beneficial for the SAR civil application [23,24]. For example, the segmentation of SAR images can help terrain mapping in debris flow rescue or urban construction design. Therefore, the segmentation of SAR images can be a basic and functional research field in SAR application.

To address the FSL challenge of SAR ATR, our key objective is to exploit the transductive generalization on unlabeled samples with an auxiliary loss serving as a regularizer. In doing so, not only the induction bias can be improved by narrowing the difference between a few samples and full samples, but also the transductive generalization can be exploited through an auxiliary loss. In this paper, we propose a semi-supervised SAR ATR framework with auxiliary segmentation (SFAS). This framework utilizes the core idea of transductive generalization to explore the relative features between unlabeled SAR samples and a few labeled SAR samples. Through auxiliary segmentation of unlabeled SAR samples and information residue loss (IRL) in training, the effectiveness of features extracted by the network is boosted, and the recognition of few labeled SAR samples is promoted greatly. In addition, accurate segmentation is also beneficial for SAR image interpretation, such as precision location or topographic mapping. The framework consists of three blocks: a shared feature extractor, a classifier to accomplish the recognition of labeled SAR target images, and a decoder to accomplish the segmentation of the unlabeled SAR target images. The contributions are as follows:

1.  We focus on the transductive generalization on unlabeled samples and proposed a semi-supervised SAR ATR algorithm with an auxiliary segmentation loss as a regularizer, which improves the inductive bias of the model and provides a new source of abundant feature information for high recognition performance of SAR ATR FSL.
2.  To explore the relative features between unlabeled SAR samples and few labeled SAR samples, we construct a training loop process (TCL) to alternately perform a semi-blind training optimization, and an information residue loss (IRL) to build a

progressive optimization of recognition and segmentation. It can help the model optimize towards information compilation of segmentation and recognition.

3. The proposed framework achieves competitive performance to state-of-the-art methods on standard benchmarks and is easy to follow. The recognition rates of 40 training samples in each class are above 95.50%, and the rates of 20 training samples in each class are above 94.00%.

The remainder of this paper is organized as follows: the framework of the proposed SFAS and detailed information about some modules are presented in Section 2. Section 3 demonstrates the evaluation of performance based on the experimental results. Finally, Section 4 draws a brief conclusion.

## 2. Proposed Method

The motivation of the proposed algorithm is to utilize unlabeled samples which are relative to a few labeled SAR samples to explore the transductive generalization from unlabeled to few labeled. Limited by the practical circumstances, SAR ATR has to face the recognition of FSL, which leads to most of the existing methods failing to learn a helpful inductive bias on a few labeled samples. Nevertheless, in reality, there are often abundant unlabeled SAR samples which have a similar distribution with few labeled samples. By exploring the transductive generalization of unlabeled samples on an effective algorithm, the recognition of FSL in SAR ATR can be promoted and achieve state-of-the-art performance.

In this section, the proposed SFAS is introduced. First, we elucidate the training loop process and framework of the SFAS. Then, the information residue loss and specific baseline of the network are described in detail. To avoid confusion, it is specifically stated here that, in our paper, the labeled samples refer to the data with class labels but without segmentation labels, while unlabeled samples refer to the data with segmentation labels but without class labels.

### 2.1. Training Loop and Framework of Proposed SFAS

To explore the transductive generalization of unlabeled samples, we focus on extracting more generalized and effective features. Thus, we proposed a training loop process to make the network learn from multiple tasks and seek more generalized features which satisfy the optimization of multi-task. The description of the novel training loop is as follows.

As shown in Figure 1, during the training, there are mainly two steps: training data generating and training loop. In the training data generating phase, the random generator generates the unlabeled and labeled set required in the training process by sampling randomly from the training set. In addition, in one training loop, there is twice semi-blind training for recognition and segmentation.

We use $D_U$ to denote the unlabeled set, $D_L$ to denote the labeled set, $M_E$, $M_D$, $M_C$ to denote the extractor, decoder, and classifier, and $P_E$, $P_D$, and $P_C$ to denote the corresponding learnable parameters. In the first semi-blind training in one training loop, the unlabeled set $D_U$ goes through extractor $M_E$ and decoder $M_D$ to calculate a segmentation loss $L_S$. The segmentation loss $L_S$ is only employed to update $P_D$, and $L_S$ are combined with recognition loss $L_R$ of the last training loop based on IRL to update $P_E$. Then, in the second semi-blind training, the few labeled set $D_L$ goes through extractor $M_E$ and decoder $M_C$ to calculate a segmentation loss $L_R$. The loss $L_R$ is only employed to update $P_C$, and $L_R$ are combined with $L_S$ based on IRL to update $P_E$ again.

By alternately performing semi-blind training for recognition and segmentation, this training loop can gradually exploit the transductive information of unlabeled samples to construct a helpful inductive bias on a few labeled samples.

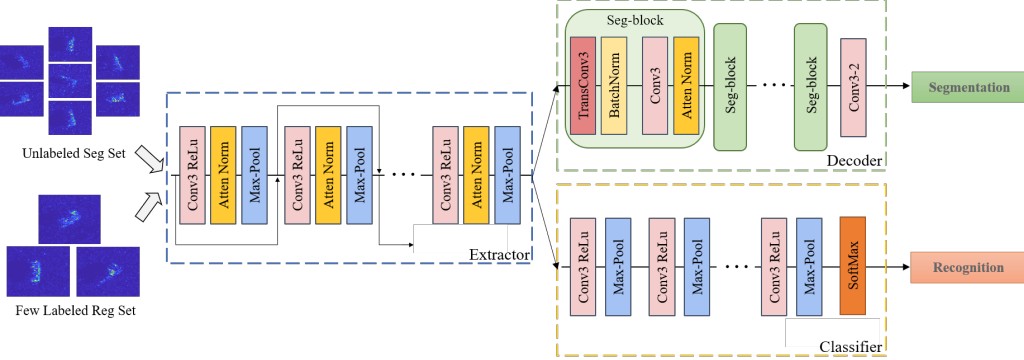

**Figure 1.** Baseline architecture.

Besides the shared feature extractor, a classifier and a decoder are shown in Figure 1, and the batch normalization and attention mechanism are employed in some level of multi-level feature extractors to ensure the network focus on the more effective features among the extracted features.

### 2.2. Information Residue Loss

The basic segmentation and recognition loss are set as the same form of cross-entropy cost. It should be noted that the design of the loss function for the classifier and decoder can promote each other and improve the accuracy simultaneously because the same form of loss can help construct the manifold structure and similar constraints. The details are as follows.

The recognition loss is set as the cross-entropy cost function, which is presented as

$$L_r(\mathbf{w}, \mathbf{b}) = -\sum_{i=1}^{C} y_i \log(p(y_i|\mathbf{x})), \tag{1}$$

where $p(y_i|\mathbf{x})$ is the probability vector of recognition result of the $i$th SAR chip, $y_i$ is the recognition labels, and $C$ is the number of the recognition classes.

The function of the segmentation loss is defined as

$$L_s(\mathbf{w}, \mathbf{b}) = -\frac{1}{n} \sum_{i=1}^{V} \mathbf{s}_i \log(m(\mathbf{s}_i|\mathbf{x})), \tag{2}$$

where $\mathbf{m}(\mathbf{s}_i|\mathbf{x})$ is the binary matrix of segmentation result of all pixel on the $i$th SAR chip, $n$ is the number of pixels in a SAR chip, $\mathbf{s}_i$ is the segmentation labels, and $V$ is the number of the segmentation types.

Then, during the proposed training loop, the recognition loss is constructed based on the proposed information residue loss by combining the segmentation loss of the last training loop. The calculation is as follows.

For the recognition loss of IRL, it is defined as:

$$L_r^t = -\sum_{i=1}^{C} y_i^t \log\left(p^t\left(y_i^t\middle|x^{(L,t)}\right)\right) - \frac{\alpha}{n} \sum_{i=1}^{V} \mathbf{s}_i^{t-1} \log\left(\mathbf{p}^{t-1}\left(\mathbf{s}_i^{t-1}\middle|x^{(L,t-1)}\right)\right), \tag{3}$$

where $t$ is the training steps, and $\alpha$ denotes the parameter to adjust the residue ratio of segmentation information of the last loop.

The computation of the parameter $\alpha$ in IRL is

$$\alpha = \frac{-1}{t} + 1. \tag{4}$$

For the segmentation loss of IRL, the function is

$$L_s^t = -\frac{1}{n} \sum_{i=1}^{V} \mathbf{s}_i^t \log\left(\mathbf{p}^t\left(\mathbf{s}_i^t \middle| x^{(L,t)}\right)\right) - \alpha \sum_{i=1}^{C} y_i^{t-1} \log\left(p^{t-1}\left(y_i^{t-1} \middle| x^{(L,t-1)}\right)\right) \tag{5}$$

The proposed novel IRL utilizes the combination of the current recognition loss and information residue of previous segmentation loss to update the extractor or vice versa. This technology, at first, can lead to more effective optimization separately for recognition and segmentation. Then, when the training is going on, the integration of the losses of both recognition and segmentation can make the optimization of the extractor towards higher accuracy of the recognition and segmentation simultaneously, and further force the network gradually extract more generalized and effective features from unlabeled and labeled samples. It is helpful to exploit the transductive generalization from unlabeled samples to construct an effective inductive bias on a few labeled SAR samples.

Therefore, the proposed SFAS not only enables few-shot learning by employing unlabeled SAR target images but also provides an effective way to apply deep learning SAR ATR methods in practical applications by acquiring a small amount of labeled SAR images and various flexible SAR images as an auxiliary task.

### 2.3. Baseline Architecture

To exploit the transductive generalization on unlabeled samples with an auxiliary loss, each part of the architecture needs to be designed carefully. To extract more generalized and effective features for recognition and segmentation, the shared feature extractor needs to face the two challenges of recognition and segmentation in the training process. The classifier and decoder for the recognition and segmentation respectively need to employ the fine-tune of downstream subtasks.

Therefore, the baseline architecture is designed as shown in Figure 1. From the shared extractor in the blue box of Figure 1, the convolutional layers are linked by the residual way, and the attention mechanism block that contains spatial and channel attention modules [25] is employed to focus on the crucial local feature and compress the useless information in each layer. In addition, the leaky ReLu and batch normalization are employed to improve the generalization of the extractor.

The decoder shown in the green box of Figure 1 is mainly constructed by trans-convolutional blocks. These blocks consist of a trans-convolutional layer to reconstruct the structure of the SAR images, and a convolutional layer with attention and batch normalization to increase the generalization. Finally, a segmentation layer (Conv3-2) outputs the segmentation results. The classifier shown in the yellow box of Figure 1 is simply stacking the convolutional layer and max-pool layer to control the parameter of the classifier, which is an effective structure of classifier in SAR ATR.

Through the method described above, the recognition of SAR ATR under limited training data has the potential to achieve high performance. The next section shows the experiments and the results.

## 3. Experimental Results

In this section, the dataset for evaluation is presented. Then, we evaluated the performance of recognition and auxiliary segmentation under different sample numbers to validate the effectiveness and the transductive information of the proposed SFAS.

### 3.1. Dataset and Network Setup

The MSTAR dataset is a benchmark dataset for the SAR ATR performance assessment. The dataset contains a series of 0.3 m × 0.3 m SAR images of ten different classes of ground targets. The optical images and corresponding SAR images of ten classes of targets in the MSTAR dataset are shown in Figure 2. The training and testing data have the same ten target classes but different depression angles. The training data are captured under

a depression angle of 17°, and the testing data are captured under that of 15°. It should be noted the ground truth of segmentation is roughly acquired by manual annotation using the tool named OpenLabeling, and there are also many available accurate automatic methods [19,20] to segment. The distribution of the training and testing images is listed in Table 1.

**Table 1.** MSTAR dataset used in experiments.

| Target Type | BMP2 | BRDM2 | BTR60 | BTR70 | D7 | 2S1 | T62 | T72 | ZIL131 | ZSU235 |
|---|---|---|---|---|---|---|---|---|---|---|
| Training (17°) | 233 | 298 | 256 | 233 | 299 | 299 | 299 | 232 | 299 | 299 |
| Testing (15°) | 195 | 274 | 195 | 196 | 274 | 274 | 273 | 196 | 274 | 274 |

The hyperparameters in the proposed network are included in Table 2. The input SAR images have the size $80 \times 80$. For each convolutional layer, the size of the stride is set as $1 \times 1$. For each max-pooling layer, the size of the stride is set as $2 \times 2$. The batch size is set as 64, and the learning rate is initialized as 0.0001. The labeled samples are based on the selected labeled training sample number in the experiments, and are randomly chosen from the whole dataset; then, the rest of the dataset is regarded as the unlabeled training samples. For example, if the number of selected labeled training samples is 60 for each class, it means that for each class, there are 60 randomly chosen labeled samples from the whole dataset, and the rest of the samples for each class are set as the unlabeled samples.

**Table 2.** Hyper-parameters of the proposed framework.

| Module | Layers | BatchNorm or Attention |
|---|---|---|
| Extractor | Conv3×3@16 | BatchNorm |
| | ReLu + MaxPool | Neither |
| | Conv3×3@32 | BatchNorm |
| | ReLU + MaxPool | Neither |
| | Conv3×3@64 | Both |
| | ReLU + MaxPool | Neither |
| Decoder | Trans-Conv3×3@64 | BatchNorm |
| | Conv3×3@32 | Both |
| | Trans-Conv3×3@32 | BatchNorm |
| | Conv3×3@16 | Both |
| | Trans-Conv3×3@16 | BatchNorm |
| | Conv3×3@8 | Neither |
| | Conv3×3@2 | Neither |
| Classifier | Conv4×4@128 | BatchNorm |
| | ReLU + MaxPool | Neither |
| | Conv4×4@256 | BatchNorm |
| | ReLU + MaxPool | Neither |
| | SoftMax | Neither |

The proposed method is tested and evaluated on a computer with Inter Core I7-9700K at 3.6 GHz CPU, Gefore GTX 1080ti GPU with two 16 GB memories. The proposed method is implemented using the open-source PyTorch framework.

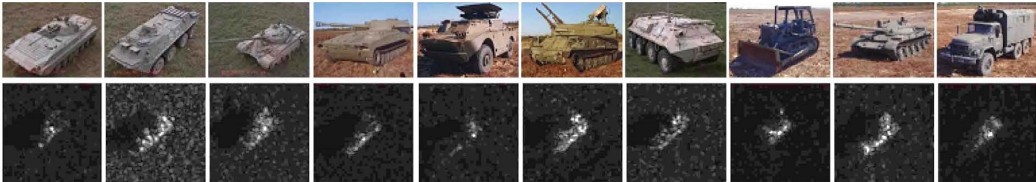

**Figure 2.** Optical images and corresponding SAR images of ten classes of objects in the MSTAR database. (From **left** to **right**: BMP2, BTR70, T72, 2S1, BRDM2, ZSU234, BTR60, D7, T62, and ZIL131).

### 3.2. Recognition Performance under SOC

The recognition results of the proposed SFAS under standard operating condition (SOC) are shown in Table 3. From the table, facing 5-shot each target without pre-training, our method can achieve an 82.06% overall recognition ratio, and when the training number of each class in few labeled SAR samples is 20 and 10 respectively, the recognition rate just decreases from 94.18% to 89.28%. It indicates that the transductive generalization on unlabeled samples can improve the recognition performance when the labeled samples decrease. When the number of each class in few labeled SAR samples is 40, it gets above 95.5%, and eight classes of all targets are above 98.00%. It shows that the inductive bias constructed by the proposed SFAS is beneficial for the recognition of few labeled samples, though the two classes of the target are more sensitive to the decreased labeled samples. In addition, it is clear that, when the number of each class in few labeled SAR samples is larger than 80, the recognition rate can get above 98.5%. It demonstrates the effectiveness of the proposed SFAS in SAR ATR under few labeled SAR samples. When all the training samples are employed, the proposed method can achieve state-of-art performance in SAR ATR.

**Table 3.** Recognition accuracy (%) of SOC under different numbers of labeled images in each classes.

| Class | Labeled Number in Each Class | | | | | | |
|---|---|---|---|---|---|---|---|
| | **5** | **10** | **20** | **40** | **80** | **100** | **All** |
| BMP2 | 38.97 | 92.31 | 98.46 | 98.97 | 97.95 | 99.49 | 100.00 |
| BTR70 | 93.80 | 98.18 | 97.45 | 100.00 | 99.49 | 98.47 | 100.00 |
| T72 | 71.79 | 73.85 | 96.41 | 99.49 | 95.92 | 100.00 | 99.49 |
| BTR60 | 100.00 | 98.47 | 96.41 | 94.36 | 98.46 | 98.97 | 97.44 |
| 2S1 | 98.91 | 86.50 | 93.80 | 81.39 | 98.54 | 98.91 | 97.81 |
| BRDM2 | 67.52 | 93.07 | 100.00 | 99.64 | 100.00 | 98.91 | 100.00 |
| D7 | 84.62 | 86.08 | 99.64 | 98.18 | 99.64 | 100.00 | 99.27 |
| T62 | 93.37 | 83.67 | 80.24 | 89.01 | 98.53 | 99.64 | 100.00 |
| ZIL131 | 86.86 | 93.43 | 91.84 | 98.54 | 97.45 | 97.81 | 100.00 |
| ZSU234 | 77.74 | 84.67 | 90.15 | 99.64 | 98.54 | 100.00 | 100.00 |
| Average | 82.06 | 89.28 | 94.18 | 95.62 | 98.52 | 99.22 | 99.42 |

### 3.3. Recognition Performance under EOCs

In the practical application, there are more complex challenges in SAR ATR. For example, the training samples and testing samples are acquired under larger depression angle differences, which leads to obvious recognition failures. Therefore, in this subsection, the recognition performance under extended operating conditions (EOCs) such as the variances of the depression angle, target configuration, and version are evaluated for the robustness and effectiveness of the proposed SFAS.

Thanks to SAR images being sensitive to the depression angle, the experiments under a larger depression angle, EOC-D, are evaluated. The training and testing samples are listed in Table 4. The training samples are captured at 17°, and the testing samples are captured at 30°. The recognition performances under EOC-D are listed in Table 5. From Table 5, it is clear that our SFAS can achieve high performance under 100, 80 and 40 training samples for each class. Furthermore, when the training samples for each class are 20 and

10, respectively, the recognition performances of EOC-D are 94.61% and 90.18%. It has illustrated that our method can handle the few-shot effect under a large depression angle.

**Table 4.** Training and testing dataset under EOCs.

| Train | Number | Test (EOC-D) | Number |
|-------|--------|--------------|--------|
| 2S1 | 299 | 2S1(b01) | 288 |
| BRDM2 | 298 | BRDM2(E71) | 287 |
| T72 | 232 | T72(A64) | 288 |
| ZSU234 | 299 | ZSU234(d08) | 288 |

| Train | Number | Test (EOC-C) | Number |
|-------|--------|--------------|--------|
| BMP2 | 233 | T72(S7) | 419 |
| BRDM2 | 298 | T72(A32) | 572 |
| BTR70 | 233 | T72(A62) | 573 |
| T72 | 232 | T72(A63) | 573 |
|  |  | T72(A64) | 573 |

| Train | Number | Test (EOC-V) | Number |
|-------|--------|--------------|--------|
| BMP2 | 233 | T72(SN812) | 426 |
|  |  | T72(A04) | 573 |
| BRDM2 | 298 | T72(A05) | 573 |
|  |  | T72(A07) | 573 |
| BTR70 | 233 | T72(A10) | 567 |
|  |  | BMP2(9566) | 428 |
| T72 | 232 | BMP2(C21) | 429 |

At the same time, the variances of target configuration and version, EOC-C and EOC-V, are evaluated for the effectiveness and practicability of our SFAS. The training and testing samples for EOC-C and EOC-V are listed in Table 4. The training samples are captured at 17°, and the testing samples for EOC-C and EOC-V are captured under 17° and 15°. The recognition performances for EOC-C and EOC-V are listed in Tables 6 and 7.

For the recognition performance of EOC-C, it is obvious that the SFAS can achieve high performance, even facing few samples for each class. When the numbers of training samples are larger than 20 for each class, the overall recognition performances are higher than 94%. In addition, the recognition ratio of 20 in each class is 94.23% and the recognition ratio of 10 in each class is 88.99%. When the number of training samples is decreasing from 100 to 20, the recognition of EOC-C is still effective. It indicated that the SFAS faced the few-shot effect and showed certain robustness under EOC-C.

**Table 5.** Recognition accuracy (%) of EOC-D under different numbers of labeled images in each class.

| | EOC-D | | | | |
|---|---|---|---|---|---|
| **Class** | **Labeled Number in Each Class** | | | | |
| | **100** | **80** | **40** | **20** | **10** |
| BRDM2 | 99.65 | 99.30 | 99.65 | 99.65 | 100.00 |
| 2S1 | 98.96 | 98.61 | 98.61 | 99.65 | 99.31 |
| T72–A64 | 92.71 | 92.36 | 87.85 | 83.33 | 73.26 |
| ZSU234 | 98.61 | 97.22 | 98.61 | 95.83 | 88.19 |
| Average | 97.48 | 96.87 | 96.18 | 94.61 | 90.18 |

**Table 6.** Recognition accuracy (%) of EOC-C under different numbers of labeled images in each class.

| Class | EOC-C | | | | |
| | Labeled Number in Each Class | | | | |
| | 100 | 80 | 40 | 20 | 10 |
|---|---|---|---|---|---|
| T72-A32 | 97.73 | 97.73 | 96.85 | 96.15 | 95.99 |
| T72-A62 | 98.95 | 97.73 | 97.03 | 94.94 | 92.32 |
| T72-A63 | 97.73 | 95.81 | 93.54 | 93.54 | 92.15 |
| T72-A64 | 93.19 | 93.19 | 92.50 | 91.97 | 86.56 |
| T72sn-s7 | 99.05 | 97.14 | 96.90 | 94.51 | 93.32 |
| Average | 97.23 | 96.27 | 95.28 | 94.21 | 92.03 |

**Table 7.** Recognition accuracy (%) of EOC-V under different numbers of labeled images in each class.

| Class | EOC-V | | | | |
| | Labeled Number in Each Class | | | | |
| | 100 | 80 | 40 | 20 | 10 |
|---|---|---|---|---|---|
| BMP2sn-9566 | 90.65 | 91.36 | 89.49 | 63.08 | 58.18 |
| BMP2sn-c21 | 90.91 | 90.68 | 90.21 | 84.38 | 56.41 |
| T72sn-812 | 98.36 | 98.83 | 98.59 | 99.77 | 99.30 |
| T72-A04 | 98.08 | 97.03 | 98.08 | 97.73 | 99.30 |
| T72-A05 | 96.68 | 95.99 | 96.68 | 95.64 | 98.78 |
| T72-A07 | 97.56 | 97.03 | 97.03 | 94.42 | 98.95 |
| T72-A10 | 98.77 | 98.59 | 98.59 | 98.24 | 98.77 |
| Average | 96.16 | 95.88 | 95.85 | 94.23 | 88.99 |

From the recognition results of EOC-V, the recognition ratios under a different number of training samples for each class are higher than 90.000%. When the number of training samples is decreasing from 100 to 10, the recognition ratio under EOC-V is decreasing slowly and smoothly, from 97.23% to 92.03%. It has illustrated that the proposed SFAS has shown its effectiveness and robustness under EOC-V.

From the recognition performance of EOCs, the effectiveness and practicability of the proposed SFAS have been evaluated. For further validation, the segmentation results are shown as follows.

### 3.4. Segmentation Performance

The overall accuracy of the segmentation can achieve 99.10% for the segmentation of target and background. The computation equation of segmentation accuracy is quoted from [26]. The partial qualitative segmentation result of the proposed SFAS is shown in Figure 3. From the three rows, it is clear that the segmentation results are accurate compared with the ground truth. A quantitative comparison regarding target accuracy, background accuracy, accuracy, and Intersection over Union (IoU) is listed in Table 8. In conclusion, the proposed SFAS can employ the transductive generalization from auxiliary segmentation to improve the recognition of few labeled samples, and utilize the recognition features to back feed the segmentation.

**Table 8.** Segmentation performance (%) of different methods.

| Methods | Target Accuracy | Background Accuracy | Accuracy | IoU |
|---|---|---|---|---|
| Otsu | 58.17% | 88.35% | 73.26% | 52.10% |
| Canny | 79.12% | 90.13% | 84.62% | 72.01% |
| Ours | 99.24% | 98.97% | 99.10% | 98.23% |

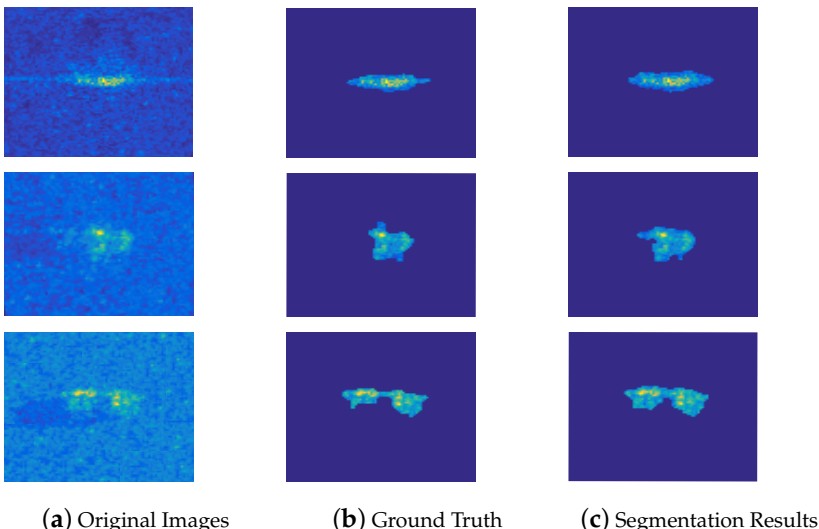

(**a**) Original Images      (**b**) Ground Truth      (**c**) Segmentation Results

**Figure 3.** Three randomly selected samples to show the segmentation performance. Each row represents one sample including original image, ground truth, and segmentation result (from **left** to **right**).

### 3.5. Comparison and Ablation

Regarding the different numbers of labeled training images, we compare our methods with other SAR ATR methods, including five traditional methods (PAC+SVM, ADaboost, LC-KSVD, DGM, and Gauss) and few-shot learning methods. Among these few-shot learning methods, there are three data augmentation methods (GAN-CNN, MGAN-CNN, and ARGN), one metric-based methods (TMDC-CNNs), and four model-based methods (DNN1, DNN2, CNN, and Semisupervised). When the number of labeled training images is 20, 40, 80, and all samples from each target, it can be seen that our proposed SFAS achieves a higher recognition rate than other methods in Table 9. Even if the number of each target in the training dataset is only 20 samples, the recognition rate of our method is above 94%, while other methods are mostly below 86%. Therefore, the results can not only validate the effectiveness of our proposed SFAS in the case of employing all training images in the dataset but also in the case of few-shot learning.

**Table 9.** Recognition accuracy (%) under different numbers of labeled images.

| Method | Labeled Number | | | | | |
|---|---|---|---|---|---|---|
| | ALL | 100 | 80 | 40 | 30 | 20 |
| PCA+SVM [27] | 94.32 | - | 92.48 | 87.95 | - | 76.43 |
| ADaboost [9] | 93.51 | - | 91.45 | 86.45 | - | 75.68 |
| LC-KSVD [9] | 95.13 | - | 93.23 | 87.39 | - | 78.83 |
| DGM [9] | 96.07 | - | 92.85 | 88.14 | - | 81.11 |
| Gauss [28] | 97.16 | - | 94.10 | 87.51 | - | 80.55 |
| DNN1 [29] | 95.54 | - | 93.04 | 86.98 | - | 77.86 |
| DNN2 [30] | 96.50 | - | 93.76 | 87.73 | - | 79.39 |
| CNN [9] | 97.03 | - | 93.88 | 88.35 | - | 81.80 |
| GAN-CNN [9] | 97.53 | - | 94.91 | 90.13 | - | 84.39 |
| MGAN-CNN [9] | 97.81 | - | 94.91 | 90.82 | - | 85.23 |
| TMDC-CNNs [31] | - | 99.09 | - | - | 86.93 | - |
| ARGN [32] | 98.00 | - | 95.98 | - | 87.20 | - |
| Semisupervised [33] | - | - | 98.65 | 97.11 | - | 92.62 |
| The Proposed | 99.42 | - | 98.52 | 95.62 | - | 94.18 |

For further automation and practicality, we utilized visual salience and morphological methods to automatically segment the target from the background, called auto-seg labels, and we replace the manual segmentation labels with the auto-seg labels to validate the effectiveness of our method. When the labeled samples are 40 and 20, the recognition ratios are 95.15% and 93.77%. It is clear that our method still achieves high performance under few-shot learning with rough auto-seg labels, which illustrated the effectiveness and practicability of our method.

The ablation experiments are also conducted shown in Table 10. When the labeled samples are 100, 80, 40, and 20, respectively, the recognition ratios without auxiliary segmentation loss are 98.93%, 95.09%, 84.62%, and 71.54% as shown in Table 10. The comparison illustrates that the segmentation loss is more effective facing few training samples.

**Table 10.** Ablation experiments of segmentation labels.

| Sample Number | 10 | 20 | 40 | 80 | 100 |
|---|---|---|---|---|---|
| With segmentation loss | 89.28 | 94.18 | 95.62 | 98.52 | 99.22 |
| Without segmentation loss | 62.57 | 71.54 | 84.62 | 95.09 | 98.93 |

### 3.6. Recognition Performance with Automatic Segmentation

In practice, samples with recognition labels may be impossible to obtain, and the segmentation label can be used to assist in recognizing small samples without recognition labels. Therefore, segmentation labels are helpful for recognition. If they are marked manually, it is really laborious, but some methods can automatically segment SAR images, which can also achieve recognition. We also conducted experiments to verify the above conclusions. We utilized visual salience and morphological methods to automatically segment the target from the background, called auto-seg labels. Then, we replace the manual segmentation labels with the auto-seg labels to validate the effectiveness of our method.

The recognition result of the proposed SFAS is shown in Table 11. The recognition ratio under a different number of training samples for each class is decreasing from 98.93% to 84.70%. When facing 100, 80 and 40 in each class, the recognition ratio achieved 98.93%, 95.67%, and 95.38%, respectively. It is clear that, with rough automatic segmentation, our SFAS still has the effectiveness and robustness of facing the few-shot effect. When the training samples for each class are 20 and 10, the overall recognition ratios with auto-segmentation are 93.19% and 84.70%. At the same time, the recognition ratios with manual segmentation are 94.18% and 89.28%. It showed that our SFAS still achieved good recognition performance and the precise segmentation can benefit the recognition. For the limited training samples, like 5-shot or 10-shot, our SFAS can employ precise segmentation from more effective segmentation methods for higher recognition ratios.

**Table 11.** Recognition accuracy (%) with automatic segmentation.

| Class | Labeled Number in Each Class | | | | |
|---|---|---|---|---|---|
| | 10 | 20 | 40 | 80 | 100 |
| BMP2 | 61.54 | 92.31 | 82.56 | 90.26 | 97.95 |
| BTR70 | 66.06 | 84.67 | 95.62 | 88.69 | 98.91 |
| T72 | 75.38 | 90.26 | 96.41 | 92.31 | 96.92 |
| BTR60 | 71.43 | 88.78 | 89.29 | 97.45 | 99.49 |
| 2S1 | 95.62 | 90.88 | 94.16 | 93.07 | 99.27 |
| BRDM2 | 94.53 | 97.08 | 92.70 | 99.27 | 98.91 |
| D7 | 84.25 | 98.53 | 100.00 | 97.07 | 98.17 |
| T62 | 97.45 | 97.96 | 99.49 | 99.49 | 100.00 |
| ZIL131 | 93.43 | 95.99 | 100.00 | 98.18 | 99.27 |
| ZSU234 | 97.81 | 94.53 | 99.64 | 100.00 | 100.00 |
| Average | 84.70 | 93.19 | 95.38 | 95.67 | 98.93 |

From the experiments with automatic segmentation, the practicability and effectiveness of our SFAS have been demonstrated. Even facing rough automatic segmentation, our SFAS still achieved high recognition performance and showed the robustness of the few-shot effect.

## 4. Conclusions

When the distributions of the few samples and complete samples have a large difference, the FSL of SAR ATR has failed to construct a helpful inductive bias. To tackle this challenge, we proposed a semi-supervised SAR ATR algorithm by employing a multi-level feature framework with auxiliary segmentation. By employing the supervised information of a few labeled samples and the transductive manifold of the unlabeled samples, the proposed novel SFAS exploits the transductive generalization to achieve better recognition performance of FSL in SAR ATR. Experimental results on the MSTAR dataset have validated the effectiveness and robustness of the proposed SFAS in few-shot recognition in SAR. The proposed SFAS not only can achieve the state of the art in recognition but also achieve superior performance in auxiliary segmentation.

**Author Contributions:** Conceptualization, C.W. and X.L.; methodology, C.W.; software, C.W.; validation, C.W., X.L. and Y.H.; formal analysis, Y.H. and J.P.; investigation, S.L., J.Y. and D.M.; resources, J.Y.; data curation, C.W. and X.L.; writing—original draft preparation, C.W. and X.L.; writing—review and editing, Y.H., J.P. and S.L.; visualization, C.W. and X.L.; supervision, J.Y. and D.M.; project administration, Y.H., J.P., J.Y. and D.M.; funding acquisition, J.Y. All authors have read and agreed to the published version of the manuscript.

**Funding:** This work was supported by the National Natural Science Foundation of China under Grants 61901091 and 61901090.

**Conflicts of Interest:** The authors declare no conflict of interest.

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
