# Peer review of "Semi-Supervised SAR ATR Framework with Transductive Auxiliary Segmentation"

_remotesensing, doi:10.3390/rs14184547_

Round 1
Reviewer 1 Report
I have read the manuscript many times and I have to admit that a lot of things are unclear to me.
Namely, all images in the dataset are segmented and labeled. Training images are selected randomly (labeled and segmented). Are all segmented images from the dataset are labeled?
According to the authors in one training loop, there is twice semi-blind training for recognition and segmentation. Does this mean that a segmented image and a labeled image of the same object type participate in one training loop? Or in one training loop a segmented image of one object and a labeled image of another object have been used?
I would like more discussion regarding Table 2. For example BTR60 achieves 100% recognition accuracy with only 5 labeled images and afterword the accuracy is dropping! There are some more similar examples within this table. That should be explained and discussed. The same is for Table 4. and Table 6.
In subsection 3.3. Recognition Performance under EOCs why in the tables all types of objects (10 of them) aren't taken into consideration?
Subsection 3.4. Segmentation Performance I can't see how the results presented in Table 7. (almost excellent segmentation results) correspond to Figure 3.! From the Figure 3. there are obvious large differences between Ground Truth and Segmentation Results.
Author Response
Thank you very much for your comments concerning our manuscript entitled “Semi-Supervised SAR ATR Framework with Transductive Auxiliary Segmentation” (remotesensing-1779645). We have revised the manuscript following your comments and suggestions. The changes made and the responses to comments are as follows.

Reviewer 2 Report
This is a nice paper. If there are no english grammar issues, it can be published.
Author Response

(The authors gave the same response as above.)

Reviewer 3 Report
In this paper, the segmentation task is used to assist small sample SAR target recognition. Alternating semi-supervised learning is proved to be valuable and effective. Regarding this paper, I have the following suggestions.
1. For the main part of the experiments in this paper, segmented samples are obtained by manual labeling. The manual segmentation samples are much more expensive to obtain than the category samples. The main value of the proposed method is that good results can still be achieved under automatic segmentation methods. I suggest to set the experimental condition for the main part of the experiment that the segmentation samples are obtained automatically.
2. As shown in Table 9 and Table 8, the proposed method is still better than the other methods even without the segmentation part when the number of samples is sufficient. This illustrates the power of the extractor in this paper. What is the effect of other methods if they use the same extractor and classifier? Ablation experiments on strong feature extractors are necessary.
3. There is a typo in line 129. The word " segmentation" should be "recognition".
4. As the main contribution, the author proposes to use a segmentation task to improve recognition performance. The segmentation subnetwork is a classical coder-decoder structure. Indeed, there are some new segmentation methods for SAR images, such as: Fast Task-Specific Region Merging for SAR Image Segmentation, Fast SAR Image Segmentation with Deep Task-Specific Superpixel Sampling and Soft Graph Convolution. The authors are suggested to give more explanation about the advantage of coder-decoder structure segmentation network, compared with the above two recent methods.
Author Response

(The authors gave the same response as above.)

Reviewer 4 Report
Please find the comments in the uploaded PDF file.

Author Response

(The authors gave the same response as above.)

Round 2
Reviewer 1 Report
I am still of the opinion the remark:
"I would like more discussion regarding Table 2. For example BTR60 achieves 100% recognition accuracy with only 5 labeled images and afterword the accuracy is dropping! There are some more similar examples within this table. That should be explained and discussed. The same is for Table 4. and Table 6. "
requires a more detailed analysis and not requires a more detailed analysis and not just a statement that it is the result of a random selection.
That could be good contribution.
Author Response
Dear Editors and Reviewers,
Thank you very much for your comments concerning our manuscript entitled “Semi-Supervised SAR ATR Framework with Transductive Auxiliary Segmentation” (remotesensing-1779645). We have revised the manuscript following your comments and suggestions. The changes made and the responses to comments are as follows.

Reviewer 4 Report
All comments from the reviewer have been solved well and the current version is worthy to be accepted.
Author Response

(The authors gave the same response as above.)
